# Tibia Valga Correction by Extraperiosteal Fibular Release in Multiple Exostosis Disease

**DOI:** 10.3390/biomedicines11102841

**Published:** 2023-10-19

**Authors:** Adyb-Adrian Khal, Emilie Peltier, Elie Choufani, Jean-Marc Guillaume, Franck Launay, Jean-Luc Jouve, Sébastien Pesenti

**Affiliations:** 1Department of Paediatric Orthopaedics, AP-HM Timone Enfants, 13005 Marseille, France; emilie.peltier@ap-hm.fr (E.P.); elie.choufani@ap-hm.fr (E.C.); jean-marc.guillaume@ap-hm.fr (J.-M.G.); franck.launay@ap-hm.fr (F.L.); jean-luc.jouve@ap-hm.fr (J.-L.J.); sebastien.pesenti@ap-hm.fr (S.P.); 2Department of Orthopaedics and Traumatology, Iuliu Hatieganu University of Medicine and Pharmacy, 400000 Cluj-Napoca, Romania

**Keywords:** exostosis, genu valgum, tibia valga, lateral release, wound healing

## Abstract

Genu valgum is a frequent deformity encountered in Multiple Hereditary Exostosis (MHE) patients. If left untreated, lower limb deformity leads to poor functional outcomes in adulthood. Our hypothesis was that in some cases, fibular shortening would lead to a lateral epiphysiodesis-like effect on the tibia. We herein report the case of a 6-year-old child with MHE who underwent extraperiosteal resection of the fibula for tibia valga correction. To obtain the lateral release of the calf skeleton, resection included inter-tibio-fibular exostosis along with proximal fibular metaphysis and diaphysis without any osseous procedure on the tibia. Gradual improvement of the valgus deformity occurred during follow-up (HKA from 165° preop to 178° at 27-month follow-up). Lateral release of the fibula led to an increase in the fibula/tibia index (from 93% preop to 96% at follow-up). Studying fibular growth in MHE patients could help understand how valgus deformity occurs in these patients. Even if encouraging, this result is just the report of a unique case. Further research and a larger series of patients are required to assess fibular release as a valuable option to treat valgus deformity in MHE.

## 1. Introduction

Multiple Hereditary Exostosis (MHE) is a rare orthopedic disorder which is responsible for abnormal cartilage and bone growth [1,2]. The incidence is between one and two cases/100,000 habitants/year and it has a clear male predilection [3]. Basic research has identified several genetical abnormalities determining the disease, such as a mutation of the EXT1 gene on chromosome 8 or EXT2 gene on chromosome 11 [1,2,3]. Heredity is present in 2/3 of cases and transmission is autosomal dominant [3].

The disease is characterized by the presence of multicentric, relatively symmetrical osteochondromas all over the skeleton [3]. Lesions present the same pathologic features as solitary osteochondromas [3]. These benign tumors occur in skeletally immature patients, usually manifest before 10 years of age and continue to develop with the growth of the child [1,2,3]. Typically, exostoses involve the bone circumferentially, mostly surrounding the metaphyseal regions, causing swelling and sometimes limiting joint motion [3]. There is a large spectrum of presentation, from no deformity to severe impairment of upper and lower extremities [3]. Their localizations in proximity of the joints may be responsible for growth disturbance, leading to limb deformities or length discrepancies [1,2]. The incidence of sarcomatous change in adult patients is low, ranging about 0.5–2% [3]. The preferred sites for the sarcomas are the trunk, limb girdles and knees. As for solitary osteochondroma, prognosis is dependent on the risk of malignant transformation [3]. 

Treatment principles are prevention and correction of deformity, shortening by removal variably combined with stapling, osteotomies and lengthening procedures [3]. If left untreated, lower limb deformity leads to poor functional outcomes in adulthood, painful mobilities, dislocations, limping or early onset arthritis [4]. Most patients do surprisingly well and have satisfactory function without surgery [3].

The most common lower limb deformity reported is the Genu valgum [5,6]. This can be caused by growth disturbance of the distal femur and proximal tibia or fibular shortening, but, the exact physio-pathological mechanism of valgus occurrence in these patients is still unclear [5,6,7,8,9]. 

In case of tibia deformity, several surgical options have been reported. Some authors described the use of temporary hemi-epiphysiodesis but with poor results when compared to idiopathic patients with frequent rebound phenomenon [10,11]. The other alternative for valgus correction is tibial osteotomy, but if performed too early before the end of growth, deformity recurrence may occur [10,11]. Also, these osteotomies are associated with a high rate of complications, such as non-union, infection or skin-related issues [12].

We hypothesize that in some cases, tibia valga is secondary to fibular shortness caused by osteochondroma developed next to the fibular physis, causing a lateral epiphysiodesis-like effect on the tibia. 

We herein report the case of a 6-year-old child with HME who underwent extraperiosteal resection of the fibula for tibia valga correction. 

## 2. Case Description

A 6-year-old boy was referred to our clinic for a follow-up of MHE. Diagnosis was confirmed 2 years before by consecutive radiographs and genetical analysis. Multiple exostoses were present in the upper and lower limbs, including distal femur and proximal tibia. The patient was asymptomatic and there was no sign of soft tissue compression by exostosis, but he had evident progressive genu valgum knee deformity. 

Radiography (Figure 1a,b) revealed a voluminous inter-tibio-fibular osteochondroma developed from fibular proximal metaphysis. The approximate dimensions of the bone tumor were about 4/3/2 cm (cranio-caudal/antero-posterior/latero-lateral). The lesion had all the characteristics of an exostosis [3]: benign-appearing bone lesion, bony excrescence with well-defined margins, thin outer cortex which flares from the host bone cortex and internal lesion in cancellous bone which blends with the cancellous bone of the metaphysis [3]. There was no sign of adjacent bone scalloping, but bowing of the tibia was present. We also performed an MRI exam which confirmed the lesion and showed close relation to the tibiofibular trunk and the anterior tibial artery, but without any sign of compression (Figure 1c). Long-leg radiographs showed valgus deformity of the tibia (HKA angle: 166°, MAD: 29 mm) (Figure 2). 

Due to the evolution of the genu valgum deformity, we decided to perform an extraperiosteal resection of the inter-tibio-fibular exostosis along to the proximal fibular metaphysis in order to obtain lateral release.

An antibiotic prophylaxis (amoxicillin/clavulanic acid 2200 mg plus gentamycin 240 mg) was given one hour prior to surgery. The surgical procedure was performed under general anesthesia, in a supine position and with a tourniquet applied at the proximal thigh. A dual medial and lateral approach of the calf were chosen. The medial approach to the proximal tibia allowed the dissection and isolation of the vessels. The skin incision begun posteriorly from the medial femoral epicondyle, about 3–4 cm over the joint line, and ended 2 cm posterior to the tibial crest, about 5 cm below the joint line. During this approach, the knee was positioned in 20°–30° flexion. After reclining the medial gastrocnemius muscle and splitting the soleus muscle, a popliteal trifurcation was identified, and the following vessels were isolated: anterior tibial artery at the inferior border of the popliteus muscle and the tibioperoneal trunk, which divided secondarily into the posterior tibial artery and peroneal artery. A lateral approach allowed for the dissection of the common fibular nerve and resection of the exostosis and the fibula. The skin incision begun 2–3 cm posteriorly from the lateral femoral epicondyle, about 2–3 cm over the joint line, and ended at the axis of the fibula at approximately 5 cm below the joint line. During this approach, the knee was positioned in full extension, but when the dissection of the common peroneal nerve started, the knee was positioned in a slight flexion of 10°–15° in order to facilitate nerve traction without any iatrogenic gesture. The common peroneal nerve was identified around the inferior border of the biceps femoris. Its course under the peroneus longus was identified, the peroneus longus tunnel was unroofed, and the nerve was mobilized posteriorly away from the proximal fibula and marked with a loop. The collateral ligament insertion as well as the biceps femoris insertion were kept in place on the fibular head. The fibular metaphysis was resected extraperiosteally from the growth physis to the proximal part of the diaphysis (Figure 3). Oscillatory saws were used to cut the bone. No osseous procedure was performed on the tibia. No perioperative complications occurred. The operating time was approximately 1.5 h and the estimated blood loss was 150 mL. Continuous suction was kept in site for 2 days and the amount of blood drainage was around 110 mL/48 h (80 mL in the first 24 h and 30 mL in the next 24 h). Regional anesthetic block was not used because we wanted to evaluate the immediate postoperative function of the common peroneal nerve.

The patient stayed in the conventional orthopedic unit for 3 days. He was discharged on the fourth day after the surgery. Total weight bearing was allowed. Crutches were used only during the first week postoperative. No cast or brace was applied. Rehabilitation was conducted regularly for 6 weeks (2 to 3 sessions/week) in order to encourage full ankle and knee mobilities and to improve balance and proprioception.

As for follow-up, the patient was evaluated every six weeks during the first three months and every six months until the first two years after surgery. The patient presented a gradual improvement of the valgus deformity. At 27 months postoperative, he was asymptomatic with a full range of motion of the knee, no clinical deformity and no knee frontal laxity. Long-leg radiographs showed the complete correction of the tibia valga (HKA: 178° AND MAD < 5 mm) (Figure 2, Table 1). Notably, due to extraperiosteal resection, there was no sign of bone union at the proximal fibula. Tibial bowing improved as well (Figure 4). Lateral release of the fibula led to an increase in the fibula/tibia index (from 93% preop to 96% at follow-up), resulting in valgus deformity improvement (Figure 5, Table 1). Using a Pearson correlation test, we found that HKA and fibula/tibia index were strongly correlated (R = 0.9390, *p* < 0.001).

We extracted clinical data from the patient’s medical charts. A descriptive study is presented, and data are presented in total frequencies and percentages. The Pearson correlation test was used to compare linear correlations between data. Statistical analyses were performed using SPSS Statistics version 26 (IBM Corp., Armonk, NY, USA). Statistical significance was set at *p* < 0.05.

## 3. Discussion

In MHE patients, osteochondromas tend to develop in proximity of joints, leading to growth disturbance and limb deformities [1,2,3,7]. There are various possibilities of clinical symptoms, from no evident deformity to severe impairment of the upper or the lower extremities [3]. Most patients do not need surgeries; they do surprisingly well and have satisfactory function [3]. But, when a growth deformity occurs, the surgical procedure must equally consist of removing the osteochondroma and correcting the defect. Left untreated, these impairments may lead to poor functional results in adulthood such as early onset arthritis or painful mobilities [4]. Usually, surgical interventions involve simple resections of the bone lesion, but, depending on the exostosis localization, additional complex dissection procedures may be required.

Genu valgum of the lower limb in these patients is frequent [1,3]. The exact physio-pathological mechanism remains unclear [1,8,13]. However, two possible scenarios may generally highlight the deformity. In some cases, abnormality in the distal femur may lead to a subsequent downstream valgus and this will also affect the knee and ankle joint alignment. On the other hand, a proximal tibia lesion would be enough to engender a “pure” valgus of the calf, without any impairment of the knee joint alignment. In both cases, the deformity is clinically evident, the ankle joint is affected, and patients may experience functional deficiencies in adulthood.

But osteochondromas can develop all over the skeleton leading to growth discrepancies [2,3], so the proximal fibula is not exempted, as was the case of our patient. In these cases, it is frequent to observe fibular shortening with no clear physio-pathological explanation [8,14,15]. On the other hand, in other benign tumors (osteoid osteomas, osteoblastomas, chondroblastomas), tumoral processes will induce local hypervascularization of the adjacent cancellous bone and this will lead to a higher blood flow in the nearby growth cartilage, resulting in a bone-lengthening effect [16,17]. According to Malhotra’s classification [9], slower growth of the fibula can hamper tibial growth on its lateral side, inducing ankle valgus. Thereby, it may produce a lateral epiphysiodesis-like effect on the tibia and, in the end, it may lead to tibia valga deformity [9]. 

Our patient had a voluminous proximal inter-tibio-fibular exostosis in close relationship with both vessels and nerves of the calf. Therefore, we chose a double approach. The lateral approach allowed for common fibular nerve control and extraperiosteal removal of the fibula metaphysis, while the medial approach allowed us to control the vessels. Although some authors consider a single lateral approach in the case of proximal fibula surgeries [18], based on our preference and experience, we preferred to avoid major vascular injury by including a medial approach. 

Exostosis removal was performed, but perhaps sole intercalary fibular removal would have been sufficient to suppress the epiphysiodesis-like effect caused by fibular shortness. This option may be considered, as the fibula is a dispensable bone and it has no biomechanical role at this site [19]. Complications following resections of the fibula are rare, but chronic pain or peroneal nerve palsy may occur [20]. In order to obtain a persistent release of fibular shortness, extraperiosteal removal is mandatory as bone union of the fibula is unwanted.

In our case, the patient presented with a valgus deformity of the left calf associated with fibula shortness. When comparing both sides, the fibula/tibia index was lower on the side of the valgus deformity. After extraperiosteal release of the fibula, the tibia shape, fibula/tibia index and, eventually, the HKA angle improved. All these modifications suggest that removal of the affected fibula metaphysis unblocked the epiphysiodesis-like effect and tibia growth resumed as normal, supporting the idea that fibular shortness is a major cause of valgus deformity onset in MHE patients.

The research community are working hard to gather the evidence for the best solutions and surgical interventions in this group of patients. In the future, we may be able to personalize the intervention and better predict outcomes in MHE patients. But future research studies should take a broader approach to the exact physio-pathological mechanism of skeletal disorder after the occurrence of osteochondromas in the proximity of joints in MHE patients. In the case of our patient, tibia valga was supposed to be a consequence of the occurrence of bone lesion, presumably impairing fibular growth and causing valgus deformity of the tibia. However, we do not know if soft tissues such as muscles or tendons could also have influenced the process of skeletal developmental impairments. The nearby femoral biceps insertion on the proximal fibula may have also played a role in the biomechanics of the knee alignment but we could not have performed a more thorough analysis. To sum up, more interdisciplinary approaches should be developed since we effectively identified an unusual case of genu valgum in an MHE patient.

Even if encouraging, this result is just the report of a unique, retrospective case report; therefore, it was subject to inherent limitations and biases. The surgical indication was not randomized, and the preference of the surgeon may have contributed to selection bias. Second, it may not be possible to judge the true incidence of complications due to the limited sample size. In addition, many potentially uncontrolled variables existed, such as the level of osteotomy or the amount of soft tissue excised. However, knee deformities in MHE may occur anytime when the distal femur or the proximal tibia are involved, and we tried to make a valid contribution to describe an unusual case of genu valgum.

## 4. Conclusions

The choice of a specific surgical procedure in the case of knee deformities in MHE is frequently based on the surgeon’s preference and experience, reflecting the lack of comprehensive studies. A novel approach of successfully treating selected genu valgum deformities is described. Studying fibular growth in MHE patients could help understanding how valgus deformity occurs in these patients. A larger series of patients with long-term follow-up is required to assess fibular release as a valuable option to treat valgus deformity in MHE.

## Figures and Tables

**Figure 1 biomedicines-11-02841-f001:**
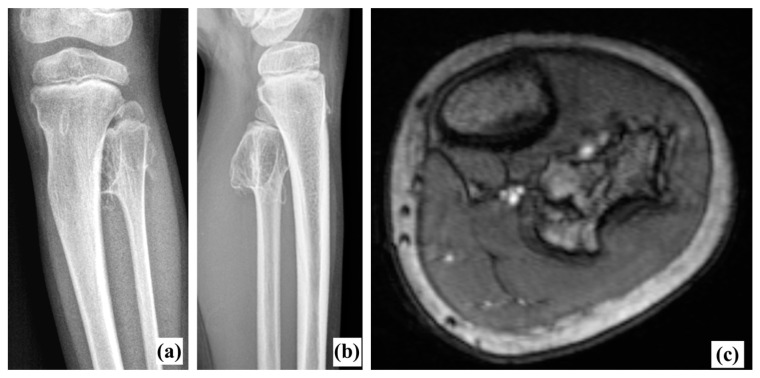
Preoperative X-rays (**a**,**b**) and MRI (**c**). Anterior tibial artery and tibiofibular trunk were immediately close to the exostosis.

**Figure 2 biomedicines-11-02841-f002:**
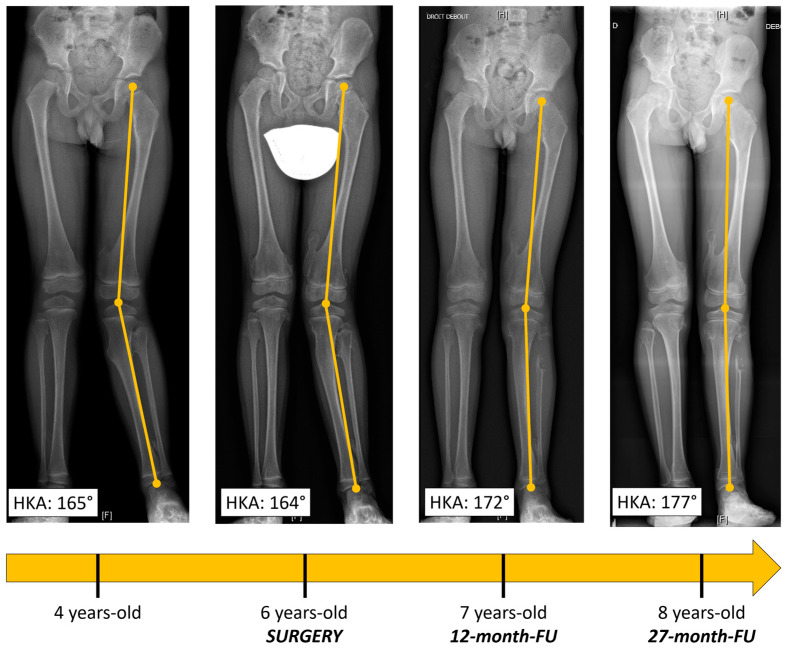
Long-leg X-rays showing gradual improvement of valgus deformity throughout follow-up.

**Figure 3 biomedicines-11-02841-f003:**
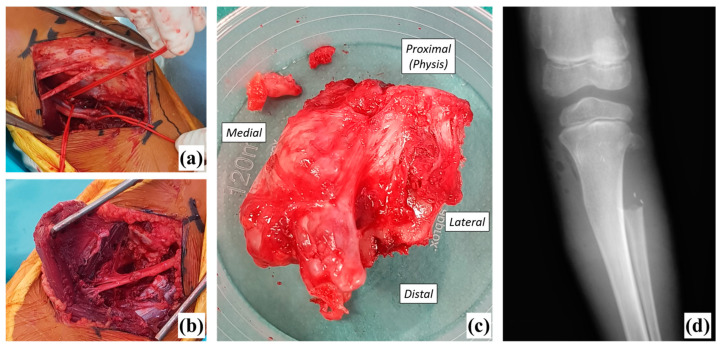
Intra-operative views ((**a**): medial approach with vessels identified; (**b**): lateral approach after resection with common fibular nerve identified and (**c**): resected exostosis with the proximal fibular metaphysis) and postoperative radiograph (**d**).

**Figure 4 biomedicines-11-02841-f004:**
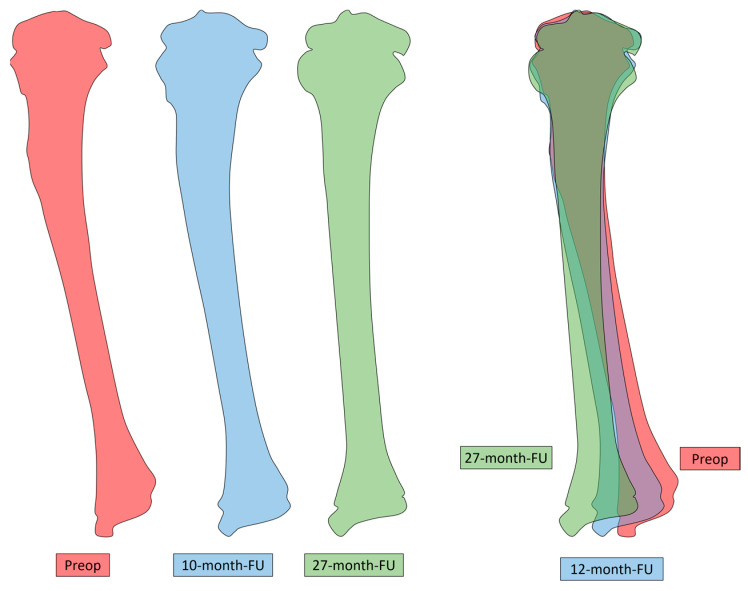
Improvement of tibia valga throughout follow-up.

**Figure 5 biomedicines-11-02841-f005:**
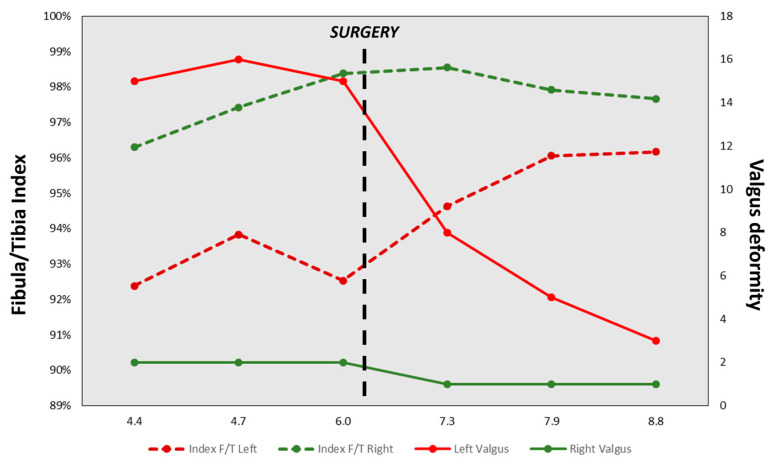
Evolution of lower leg parameters during follow-up.

**Table 1 biomedicines-11-02841-t001:** Long-leg radiographs analysis.

Age	4.4	4.7	6.0	7.3	7.9	8.8
Left Fibula (mm)	194	213	223	247	268	276
Left Tibia (mm)	210	227	241	261	279	287
Right Fibula (mm)	208	227	243	272	282	293
Right Tibia (mm)	216	233	247	276	288	300
Index F/T Left	92%	94%	93%	95%	96%	96%
Index F/T Right	96%	97%	98%	99%	98%	98%
Left Valgus (°)	15	16	15	8	5	3
Right Valgus (°)	2	2	2	1	1	1

## Data Availability

On request from the corresponding author, the data are not publicly available due to privacy and ethical reasons.

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
