# Peer review of "Tibia Valga Correction by Extraperiosteal Fibular Release in Multiple Exostosis Disease"

_biomedicines, 2023, doi:10.3390/biomedicines11102841_

Round 1
Reviewer 1 Report
You cannot discuss your so called "hypothesis" on case report, since it is not enough to either confirm or reject the hypothesis. However, case is really good explained and documented, thus bringing significant novelty to the field. Future plans and potential study designs should be additionally discussed in discussion paragraph. After this minor modifications I would recommend this paper for publication.
You cannot discuss your so called "hypothesis" on case report, since it is not enough to either confirm or reject the hypothesis. However, case is really good explained and documented, thus bringing significant novelty to the field. Future plans and potential study designs should be additionally discussed in discussion paragraph. After this minor modifications I would recommend this paper for publication.
Author Response
Thank you for giving us the opportunity to submit the revised draft of the Manuscript biomedicines-2638265, entitled „ Tibia valga correction by extraperiosteal fibular release in multiple exostosis disease” for publication in the journal of Biomedicines. We appreciate the time and effort that you dedicated to providing feedback on our manuscript and are grateful for the insightful comments on and valuable improvements to our paper. We have incorporated the suggestions made by the reviewer. Those changes are highlighted in yellow in the manuscript. Please see below, for a point-by point response to the comments and concerns.
- Thank you for this precious observation. We added supplementary information about future research plans and study designs.
Lines 168-180: “The research community are working hard to gather the evidence for the best solutions and surgical interventions in this group of patients. In the future, we may be able to personalize the intervention and better predict outcomes in MHE patients. But, future research studies should take a broader approach to the exact physio-pathological mechanism of skeletal disorders after the occurrence of osteochondromas in the proximity of joints in MHE patients. In the case of our patient, tibia valga was supposed to be a consequence of the bone lesion occurrence, presumably impairing fibular growth and causing valgus deformity of the tibia. However, we do not know if soft tissues such as muscles or tendons could also influence the process of skeletal developmental impairments. The nearby femoral biceps insertion on the proximal fibula may have also played a role in the biomechanics of the knee alignment but we could not have performed a more thoroughly analysis. To sum up, more interdisciplinary approaches should be developed since we effectively identified an unusual case of genu valgum in MHE patients.
Reviewer 2 Report
The research aim is to present a case of MHE in a child who underwent extraperiosteal resection of the fibula for tibia valga correction. The authors suggest tibia valga could be secondary to filbular shortness. It is an interesting topic.
The abstract is well written.
The introduction transposes the research into the topic and formulates the objective of the study at the end.
The case is clearly presented. The statistical software used should be stated.
The discussions interpret the cases and relate them to other results from the scientific literature. Limitations of the study are presented at the end of the section. Future research directions on this topic could be also stated.
The conclusions are concise and clear.
The references are adequate but should be edited according to the criteria of the journal.
Author Response
Thank you for giving us the opportunity to submit the revised draft of the Manuscript biomedicines-2638265, entitled „ Tibia valga correction by extraperiosteal fibular release in multiple exostosis disease” for publication in the journal of Biomedicines. We appreciate the time and effort that you dedicated to providing feedback on our manuscript and are grateful for the insightful comments on and valuable improvements to our paper. We have incorporated the suggestions made by the reviewer. Those changes are highlighted in yellow in the manuscript. Please see below, for a point-by point response to the comments and concerns.
- Thank you for this precious observation. We inserted a short paragraph on the statistical software and analysis.
Lines 118 – 122: “We extracted clinical data from patient medical charts. A descriptive study is presented and data are presented in total frequencies and percentages. The Pearson correlation test was used to compare linear correlation between data. Statistical analyses were performed using SPSS Statistics version 26 (IBM Corp., Armonk, NY, USA). Statistical significance was set at p < 0.05.”
- Thank you for this precious observation. We added supplementary information about future research plans and study designs.
Lines 168-180: “The research community are working hard to gather the evidence for the best solutions and surgical interventions in this group of patients. In the future, we may be able to personalize the intervention and better predict outcomes in MHE patients. But, future research studies should take a broader approach to the exact physio-pathological mechanism of skeletal disorders after the occurrence of osteochondromas in the proximity of joints in MHE patients. In the case of our patient, tibia valga was supposed to be a consequence of the bone lesion occurrence, presumably impairing fibular growth and causing valgus deformity of the tibia. However, we do not know if soft tissues such as muscles or tendons could also influence the process of skeletal developmental impairments. The nearby femoral biceps insertion on the proximal fibula may have also played a role in the biomechanics of the knee alignment but we could not have performed a more thoroughly analysis. To sum up, more interdisciplinary approaches should be developed since we effectively identified an unusual case of genu valgum in MHE patients.